# Psychometric Properties of the Polish Version of the 36-Item WHODAS 2.0 in Patients with Low Back Pain

**DOI:** 10.3390/ijerph17197284

**Published:** 2020-10-06

**Authors:** Agnieszka Ćwirlej-Sozańska, Agnieszka Bejer, Agnieszka Wiśniowska-Szurlej, Anna Wilmowska-Pietruszyńska, Alessandro de Sire, Renata Spalek, Bernard Sozański

**Affiliations:** 1Institute of Health Sciences, College of Medical Sciences of the University of Rzeszow, University of Rzeszow, Rejtana16C, 35-959 Rzeszow, Poland; agnbej@wp.pl (A.B.); wisniowska@vp.pl (A.W.-S.); benieks@poczta.onet.pl (B.S.); 2Faculty of Medicine, Lazarski University, Świeradowska Street 43, 02-662 Warsaw, Poland; anna.wilmowska@gmail.com; 3Physical and Rehabilitative Medicine, Department of Health Sciences, University of Eastern Piedmont, Viale Piazza D’Armi 1, 28100 Novara, Italy; alessandro.desire@gmail.com; 4Department of Geriatrics, Neurosciences, Orthopedics, Center for Geriatric Medicine (CEMI), Institute of Internal Medicine and Geriatrics, Catholic University of the Sacred Heart, L.go F.Vito 8, 00168 Rome, Italy; renata.spalek@unicatt.it or; 5Rehabilitation Unit, ‘Mons. L. Novarese’ Hospital, Str. Sotto Cerca, 13040 Vercelli, Italy

**Keywords:** ICF, disability, low back pain, rehabilitation, validity, reliability

## Abstract

The World Health Organization Disability Assessment Schedule 2.0 (WHODAS 2.0) is considered by the World Health Organization (WHO) to be a useful tool for assessing the functioning and disability of the general population as well as the effectiveness of the applied interventions. Until this study, no data regarding the validity of the 36-item WHODAS 2.0 in chronic low back pain (LBP) in Poland have been explored. This study was conducted on 92 patients suffering from chronic LBP admitted to the rehabilitation ward. The Polish version of the 36-item WHODAS 2.0, the Sf-36 Health Survey (SF-36), the Oswestry Disability Index (ODI), the Hospital Anxiety and Depression Scale (HADS) and the Visual Analogue Scale (VAS) questionnaires were applied to assess patients. The scale score reliability of the entire tool for the study population was very high. The Cronbach’s alpha test result for the entire scale was 0.92. For the overall result of the WHODAS 2.0, the Intraclass Correlation Coefficient (ICC_1,2_) was 0.928, which confirmed that the scale was consistent over time. The total result and the vast majority of domains of the 36-item WHODAS 2.0 correlated negatively with domains of the SF-36 questionnaire; thus, a higher WHODAS 2.0 score was associated with a lower score on the SF-36 questionnaire. We found that the minimal clinically important difference (MCID) for the total WHODAS 2.0 score in patients after rehabilitation for LBP was 4.87. Overall, the results indicated that the Polish version of the 36-item WHODAS is suitable for assessing health and disability status in patients with LBP.

## 1. Introduction

Currently, low back pain (LBP) is the most common health problem [1], affecting an estimated 70% to 85% of the population [2,3]. The frequency of LBP problems is still increasing [4]. LBP puts significant limitations on the functioning of individuals [5]. Chronic LBP is a serious burden on health, social, and work systems [6,7]. LBP is a major contributor to disability worldwide [8] and is in sixth place in terms of overall disease burden [9].

The World Health Organization (WHO) defines disability as a difficulty in the functioning at the level of the body, activity, or participation in social life, as experienced by a person experiencing a health problem in interaction with contextual factors [10]. The biopsychosocial conceptual model has been described in WHO’s International Classification of Functioning, Disability, and Health (ICF) [11]. In this context, the ICF might be considered to be the ideal conceptual framework to adequately describe and measure the impairments, activity limitations, and participation restrictions in people affected by LBP [12]. Indeed, a recent Italian multicenter cross-sectional study reported that back pain was the most common disease (9.6%) and the sensation of pain (ICF b280) was the most common alerted ICF item (76.3%) in a cohort of 864 patients [13].

The most common tools that have been used to measure disability in patients affected by LBP are the Oswestry Disability Index (ODI) and the Roland–Morris Disability Questionnaire [14]. However, neither of them was developed with the ICF biopsychosocial conceptual model [15]. In order to get a broader perspective of the problems experienced by people with LBP, the use of the World Health Organization Disability Assessment Schedule 2.0 (WHODAS 2.0) instrument based on the ICF may be of value [16]. A comprehensive disability assessment is important for identifying problems and needs for planning health care and rehabilitation services and for assessing the results and effectiveness of interventions [17,18].

The 36-item WHODAS 2.0 has been validated in some populations where some psychological and physical disease states occur [19]. Previous studies investigated the psychometric properties of the WHODAS 2.0, including arthritis [20], stroke [21], systemic sclerosis [22], psychotic disorders [23] or depression, and back pain [24]. Lee et al. suggested that WHODAS 2.0 may be a useful measure for patients with LBP [25]. The scale was translated and validated among patients with musculoskeletal problems in various countries, including Portugal [26], Finland [27] and Turkey [28]. The WHODAS 2.0 might be among the best measures for assessing LBP disability, being the most supportive of the biopsychosocial disability model [29]. Moreover, the Measuring Health and Disability in Europe: Supporting policy development - MHADIE, an observational, longitudinal, multicentric study conducted in seven European centers (Czech Republic, Germany, Italy, Slovenia and Spain) indicated that WHODAS 2.0 conformed good metric properties in clinical and rehabilitation samples in patients with chronic diseases including LBP [30]. Results from the World Health Organization (WHO) study on global ageing and adult health (SAGE) suggested that comparable evidence on back pain and its impact on disability is needed and internationally important so that governments can invest in rehabilitation to reduce the growing economic and social burden resulting from disability [31].

However, there are no data regarding the validity of the 36-item WHODAS 2.0 in chronic low back pain in Poland. Therefore, the present study aimed to investigate the psychometric properties of the Polish version of the 36-item WHODAS 2.0 in patients with chronic LBP.

## 2. Materials and Methods

### 2.1. Study Design and Data Collection

This study was conducted on patients with chronic LBP admitted to the rehabilitation ward of the Holy Family Specialist Hospital, Rudna Mała, Poland.

Inclusion criteria were admission to the rehabilitation ward for LBP during the study period; suffering from LBP for at least 12 weeks; aged ≥50 years. The exclusion criteria were severe neurological disorders of the central nervous system (stroke and traumatic brain injury), unstable cardiovascular diseases, active cancer, amputations and lack of consent by the patient.

All consecutively admitted patients with chronic LBP from June 2019 to March 2020 meeting the inclusion criteria were qualified for the study. The study was conducted by properly prepared and trained physiotherapists in the rehabilitation ward. The investigation was performed using a direct interview implementing a pen and paper interview method. The examination was carried out three times: study I—in the first day after hospital admission using the following tools: the 36-item WHODAS 2.0, the SF-36 Health Survey (SF-36), ODI, the Hospital Anxiety and Depression Scale (HADS), the Visual Analogue Scale (VAS), and sociodemographic data were also collected; study II—mainly two days after the study I (using the WHODAS 2.0) and study III–1 month after the completion of rehabilitation in the hospital (to assess responsiveness, the WHODAS 2.0 and VAS were used) (Figure 1). 

### 2.2. Ethics Approval

This study was approved by the Bioethics Committee of the University of Rzeszów (Resolution No. 33/05/2019). All participants were instructed on the purpose and course of the study. They also received information that they could withdraw from participation at any time without any consequences. Moreover, they were asked to sign informed consent in order to take part in the research.

### 2.3. Measurements

#### 2.3.1. The 36-item WHODAS 2.0

In accordance with the WHO rules, the 36-item WHODAS 2.0 was translated and culturally adapted by the ICF Council at the Poland Health Protection IT Systems, led by Professor Anna Wilmowska-Pietruszyńska, based on the agreement with the WHO [32].

The 36-item WHODAS 2.0 is used to measure general disability and disability in six domains: Do1 Cognition (6 items), Do2 Mobility (5 items), Do3 Self-care (4 items), Do4 Getting along (5 items), Do5 Life activities (8 items), and Do6 Participation (8 items). During the interview, the response refers to the last 30 days. Answers to the questions are rated on a 5-point scale identifying the level of difficulty or problem (1 = none; 2 = mild; 3 = moderate; 4 = severe; 5 = extreme or cannot to do). The obtained results are converted on the scale from 0 to 100 [33]. The psychometric properties of the 36-item WHODAS 2.0 have been examined in a cross-sectional study of older people in Poland [34].

#### 2.3.2. The SF-36 Health Survey (SF-36)

The SF-36 version 2.0 is a general tool for measuring the health-related quality of life. The questionnaire contains 36 items used to measure eight domains: Physical functioning (10 items), Role limitations due to physical health (4 items), Bodily pain (2 items), General health perceptions (5 items), Vitality (4 items), Social functioning (2 items), Role limitations due to emotional problems (3 items), Mental health (5 items), Reported health transition (1 item).

Additionally, changes in general health over the preceding year are recorded. The remaining items in the questionnaire concern the experiences from the preceding month. In addition, the first four domains form the Physical component scale (PCS), whereas the next four ones create the Mental component scale (MCS).

The answers given by the participants are normalized so that the quality of life measures calculated on this basis are in the range of 0–100 points, where the value 0 is always the worst and the value 100 points relates to the best quality of life [35,36]. License agreement number QM030224 was obtained for using the SF-36 v. 2.0 questionnaire for the research.

#### 2.3.3. The Oswestry Disability Index (ODI)

The modified ODI is a tool assessing the functional disability of a patient with LBP. It includes 10 items referring to pain and activities of daily living, each scored from 0 to 5. The total score is calculated through multiplying the sum, giving a range of 0 to 50 [37,38].

#### 2.3.4. The Hospital Anxiety and Depression Scale (HADS)

The HADS is a tool commonly used for self-assessment detecting non-physical symptoms of anxiety and depression. It includes 14 items, i.e., seven items refer to the anxiety subscale (HADS anxiety) and seven other items refer to the depression subscale (HADS depression) [39]. Each item is rated on a 4-point scale ranging from 3 to 0. After adjusting for six items that are scored reversely, the sum of all responses is used to calculate the two subscales [40,41].

#### 2.3.5. The Visual Analogue Scale (VAS)

The VAS is used to assess the intensity of pain using a visual scale, where 0 represents the total absence of pain, and 10 indicates unbearable pain.

#### 2.3.6. Sociodemographic Data

Sociodemographic data were collected to provide basic information concerning sex, age, place of residence and education.

### 2.4. Statistical Analyzes

In order to receive the results, it is necessary to perform statistical analyses. The obtained data were analyzed using the R software, version 3.6.1. For the initial data analysis, the researchers used descriptive statistics measures.

#### 2.4.1. Reliability Analysis

##### Internal Consistency

In order to assess the internal consistency, the Cronbach’s alpha-coefficient was used. Cronbach’s alpha values between 0.70 and <0.95 indicated the adequate internal consistency reliability of the scale [42,43].

##### Test–Retest Reliability and Measurement Error

The reliability of the 36-item WHODAS 2.0 was assessed using the test–retest method. The time between the two measurements made by different interviewers amounted to 2 days on average. During this period there should have not been significant changes in the phenomenon under study. The Intraclass Correlation Coefficient (ICC_2,1_), with a 95% confidence interval (CI), was used to measure the relative reliability [42,44]. It is the ability of a questionnaire to capture similar scores on 2 separate occasions of test administration, given when the patient’s condition has not changed [45]. The relative reliability indicates the degree of consistency and agreement between two measures [46]. The standard error of measurement (SEM) quantifies what was assessed to measure the absolute reliability. The determination of the absolute reliability of measures is critical to ensure repeated measurements with satisfactory stability and sensitivity to real changes over time [45]. The absolute reliability indicates how much dispersion and error this measurement contains [46]. As for the discussed study, the SEM was calculated as follows: SEM = SD √(1 − ICC_2,1_) [44,47,48,49]. In addition, the minimal detectable change at the level 95% (MDC_95_) was calculated. The MDC estimates the minimal amount of change in the score that confirms that the change is truly eliminating measurement error. In this case, the MDC was obtained using the formula: MDC = SEM × 1.96 × √2, where 1.96 was derived from the 0.95% CI of no change, and √2 showed two measurements assessing the change [44,47,50].

##### Internal Structure

The internal structure of the 36-item WHODAS 2.0 was also assessed by analyzing the correlations between the items in a given domain and the domain itself, as well as correlations appearing between the items and the overall result. Pearson’s correlation coefficient was used.

#### 2.4.2. Floor and Ceiling Effects

To detect the floor and ceiling effects, they were established by determining the percentage of subjects who scored the lowest or highest results with reference to the 36-item WHODAS 2.0. Floor or ceiling effects was observed if there were more than 15% of participants providing the lowest or highest possible score, respectively [51].

#### 2.4.3. Validity

##### Convergent Validity

The convergent validity was assessed by correlating the results of the 36-item WHODAS 2.0 and the SF-36 questionnaire, the HADS and the ODI. The analysis was performed by examining Pearson’s correlation coefficient. Adults with a lower quality of life and lower mood should have a higher level of disability [52]. We also assumed that the disability assessment using the ODI questionnaire would correlate with the assessment using the 36-item WHODAS 2.0. The greater the disability in the assessment of the ODI questionnaire, the higher the disability in the 36-item WHODAS 2.0.

##### Known Group Validity

The known group validity was assessed to test whether the 36-item WHODAS 2.0 distinguished two groups which should have different levels of construct. We took into account the simplicity in assessing these problems and the possibility of assigning function problems to the ICF framework. The occurrence of pain (ICF b280, a sensation of pain) was considered a health problem affecting the disability. The pain level was assessed using the VAS scale. For the purposes of the analysis, a dichotomous variable was created to divide the studied population into groups based on the following cutoffs: VAS ≤ 5 and VAS ≥ 6. Adults with higher levels of pain should be characterized by a higher level of disability [53,54]. The comparison of the 36-item WHODAS 2.0 results in the two groups was performed using the Student’s *t*-test.

#### 2.4.4. Responsiveness

The responsiveness refers to the ability of an instrument to distinguish clinically important changes as the result of an intervention. In order to assess the responsiveness, standard effect size (ES) and standardized response mean were calculated (SRM). ES is defined as a change in the mean score of the 36-item WHODAS 2.0 (between test 1 and 3) divided by the SD of the baseline score. Paired-samples *t*-test was used to examine the mean change between test 1 and test 3. SRM was calculated by dividing the mean score change by the SD of that score change. Absolute values of 0.20 or less, 0.21–0.79, and 0.80 or greater represent small, moderate, and large responsiveness, respectively, for ES and SRM [55].

To access responsiveness, the minimal clinically important difference (MCID) with its standard error (SE) was assessed [56]. The MCID was calculated on the basis of linear regression analysis, where the dependent variable was the change between 1st and the 3rd study in the 36-item WHODAS 2.0 (separately for each domain), and independent variable—change by 1 point on the VAS.

## 3. Results

### 3.1. Socio-Demographic Characteristics

In the studied adult population, 61.96% were women. The average age was 66.0 (SD = 11.6) years. Slightly more of the respondents lived in the countryside (52.17%). Most of the respondents had secondary education (48.91%). The average level of pain on the VAS scale in the studied population was 5.77 points. According to the 36-item WHODAS 2.0, the average disability score for the study group was 41.53 ± 13.84. The highest average level of disability was observed in Do2 Mobility (65.08 ± 20.49) and Do5 Life activities (60.43 ± 21.83). Regarding the quality of life, the respondents rated the worst functioning in the domains: Physical functioning, Role physical and body pain, respectively: 35.11 ± 20.91; 36.07 ± 19.87; 35.16 ± 17.06. According the ODI, the average degree of disability caused by lower back pain was 29.57 ± 6.40. The average HADS subscales anxiety and depression were 7.67 ± 3.63 and 5.51 ± 3.14 (Table 1), respectively.

### 3.2. Reliability Analysis

#### 3.2.1. Internal Consistency

The scale score reliability of the 36-item WHODAS 2.0 for the study population was very high. The Cronbach’s alpha test result for the entire scale was 0.921. As for the Cronbach’s alpha for the individual domains, it ranged from 0.786 (Do4 Getting along) to 0.904 (Do5 Life activities) (Table 2).

#### 3.2.2. Test–Retest Reliability and Measurement Error

The value of the ICC_2,1_ ranged from very high (for Do4, ICC_2,1_ was 0.936) to high (for Do5, ICC_2,1_ was 0.759). For the overall result of the WHODAS 2.0, the ICC_2,1_ was 0.928, which confirmed that the scale was consistent over an approximate 2 day period (Table 2). The total score WHODAS 2.0 result was characterized by a low measurement error (SEM = 3.77). The smallest SEM was found in the Do4 Getting along domain (SEM = 4.44), and the largest in Do5 Life activities (SEM = 11.01). The SEM for the overall result and all WHODAS 2.0 domains were less than 50% of the respective standard deviation values, except for Do3 Self-care and Do5 Life activities, where the measurement error was recorded at the limit of 50% of the standard deviation values for these domains. The MDC was the best (the smallest) for the total result (MDC_95_ = 10.45), indicating that 10.45 was the minimal amount of change in the score of an instrument that must occur for an individual in order to be sure that the change in the score is not simply attributable to measurement error (Table 2).

#### 3.2.3. Internal Structure

All subscales for WHODAS 2.0 were moderately strongly or strongly correlated with the total result (r ranged from 0.482 to 0.762) (Table 3).

### 3.3. Floor and Ceiling Effects

No floor or ceiling effects for the overall result the 36-item WHODAS 2.0 were found. However, over 15% of respondents reported the lowest possible score for WHODAS 2.0 Do1 cognition and Do4 getting along, indicating possible floor effects for these two domains (Table 2).

### 3.4. Validity

#### 3.4.1. Convergent Validity

The convergent validity was tested by correlating the results obtained with the 36-item WHODAS 2.0, the results of the SF-36, the HADS and the ODI questionnaires.

The total result and the vast majority of domains of the 36-item WHODAS 2.0 were negatively correlated with domains of the SF-36 questionnaire; thus, a higher score on the WHODAS (higher disability) was associated with a lower score on the SF-36 questionnaire (lower quality of life). The weakest correlation was found with Do3 Self-care and Do4 Getting along.

The total result and all the domains of the 36-item WHODAS 2.0 were positively correlated with each domain of the HADS questionnaire; thus, a higher score on the 36-item WHODAS 2.0 (higher disability) was associated with a higher score on the HADS questionnaire (anxiety and depression). These findings confirm that adults with higher anxiety and depression are characterized by a higher level of disability (Table 4).

The total result of the 36-item WHODAS 2.0 and all the domains were correlated with the ODI questionnaire. These findings confirming that adults with a higher physical disability measured by the ODI have a higher level of disability by the 36-item WHODAS 2.0 (Table 4).

#### 3.4.2. Known Group Validity

We found significant differences among the selected subgroups of pain. With the possible exception of Do4 Getting along, we found differences between the selected subgroups and the total score of the 36-item WHODAS 2.0. These findings indicate that adults with higher levels of pain are likely characterized by a higher level of disability (Table 5).

### 3.5. Responsiveness

The statistical evidence indicates that all WHODAS 2.0 domains decreased between the first and third study (i.e., from test 1 to test 3). Nearly all domains showed a moderate to large degree of responsiveness, respectively, as signified by the ES and SRM values. The largest MCID was found in the case of Do2 Mobility (7.93 ± 0.70), and the smallest in the case of Do1 Cognition (1.71 ± 0.34) (Table 6).

## 4. Discussion

To the best of our knowledge, this is the first study in which the researchers have evaluated the psychometric properties and validation of the Polish version of the 36-item WHODAS 2.0 engaging the patients with chronic LBP. This study is important due to the need for the implementation of valuable and reliable clinical tools for assessing the functioning and disability of patients with musculoskeletal pathology and for assessing rehabilitation progress. The 36-item WHODAS 2.0 implementation in Poland is associated with the simultaneous implementation of ICF. Indeed, in the recent past, the LBP Core Set Self-Report Checklist (LBP-CS-SRC) has been recently developed to facilitate people in self-rating activity limitations and participation restrictions [57]. Albeit LBP-CS-SRC is useful to understand the patients’ perspectives [58], the 36-item WHODAS 2.0 has been defined as an instrumental tool for the clinical assessment of disability and the ability to function in patients with LBP by the latest WHO resolution for the International Classification of Diseases 11^th^ Revision (ICD-11) [59].

The results of our research have shown that the Polish version of the 36-item WHODAS 2.0 presents good psychometric properties and can be useful for the clinical examination of adult patients with LBP in Poland.

We found a very good scale score reliability of the entire Polish version of the 36-item WHODAS 2.0. In our case, the Cronbach’s alpha test value for the whole scale was 0.92. The Cronbach’s alpha value for individual domains ranged from 0.79 to 0.90. The tool is reliable and not “redundant” i.e., it does not contain too many questions still exploring the same subject. The tool meets Nunnally’s criteria, according to which for a good Cronbach’s alpha scale it must be >0.70 [43]. Similar reliability of the WHODAS 2.0 test was received by Silva et al. while validating the Portuguese version of the 36-item WHODAS among 204 patients with musculoskeletal pain [26]. The authors of this article have confirmed the reliability of the WHODAS 2.0 by also examining 60–70 year-olds living in Poland, establishing Cronbach’s alpha for the whole scale on the level 0.89 and for individual domains it ranged from 0.85 to 0.86 [34]. Moreover, the reliability of the WHODAS 2.0 test was correspondingly obtained by Moen et al., where the Cronbach’s alpha was 0.93 for the whole score and for individual domains it ranged from 0.75 to 0.94 [60]. It is worth mentioning that other authors also obtained high WHODAS 2.0 reliability scores [61,62,63].

We confirmed the good repeatability of the 36-item Polish WHODAS 2.0. For the overall result the 36-item WHODAS 2.0, the ICC_2,1_ was 0.93 and for domains it ranged from 0.76 to 0.94. Kutlay et al. examined patients with osteoarthritis who received the ICC retest–test value for the overall score of 0.97, and for individual domains in the range of 0.87–097 [64]. As for the Chinese version of the WHODAS 2.0, the ICC values for the total score was 0.80 and for domains it ranged 0.83–0.89 [65]. Moen et al., examining patients referred for somatic rehabilitation, found an acceptable ICC reproducibility of the total score and the different domains except for self-care [60].

In our study, the SEM for the overall score and the WHODAS 2.0 domains were less than 50% of the respective standard deviation values. The exceptions were the domains of Do3 self-care and Do5 life ctivities, for which the measurement error was found to be at the level of 50% of the standard deviation value. For the overall result of the 36-item WHODAS 2.0, the SEM was 3.77, while the MCD_95_ was 10.45. In the studies of Serrano-Dueñas et al., the SEM for the WHODAS overall result slightly exceeded 50% of the measurement error (SEM = 51.7%) [66]. Silva et al., in their research, received the relatively small deviations of the SEM (2.94) and the MDC (8.15) indicated the good reliability of the 36-item WHODAS 2.0 summary score [26].

No floor effect and ceiling for the overall result of the 36-item WHODAS 2.0 was found. However, the Do1 Cognition and Do4 Getting along domains showed floor effects over 15%. The result for Do4 was actually on the border of the adopted norm. In contrast, due to the attributable proportion of patients with no cognitive problems, a high floor result was expected in the cognition domain. The low percentage of ceiling and floor scores obtained in the summary score and in the other domains could support the use of these scores in rehabilitation assessment in patients with LBP. Serrano-Dueñas et al., assessing patients with Parkinson’s disease, found floor effects in the Do1 Cognition domain, and the Do5 Life activities domain of the WHODAS 2.0 scale receiving 17.2% and 22.9%, respectively [66].

We found a positive correlation between the six domains of the Polish version of the 36-item WHODAS 2.0 (*p* < 0.001). A good correlation coefficient was identified between the total score and each domain (r = 0.482–0.762), demonstrating a good internal structure. Similarly, the total score had a good correlation with the six domains of the 36-item WHODAS 2.0 in the case of the traditional Chinese version (*p* < 0.05; r = 0.7–0.76) [67].

We also tested the convergent validity of the 36-item WHODAS 2.0. The overall result and the vast majority of domains of the 36-item WHODAS 2.0 were negatively correlated with domains of the SF-36 questionnaire; hence, a higher score on the WHODAS (higher disability) was associated with a lower score on the SF-36 questionnaire (i.e., lower quality of life). All WHODAS 2.0 domains were correlated with SF-36 domains like physical functioning, body pain, PCS and MCS. The weakest correlation was found between Do3 Self-care and Do4 Getting along of the WHODAS 2.0 questionnaire, and other SF-36 domains. Baron et al. demonstrated the strong correlation of WHODAS 2.0’s total score with the SF-36 PCS (τ = −0.51, *p* < 0.001) and moderate correlation with the SF-36 MCS (τ = −0.43, *p* < 0.001). The WHODAS 2.0 domains, likewise, were all moderately to strongly correlated with the subscale domains and total scores of the SF-36 [20]. Other authors also obtained correlations between the 36-item WHODAS 2.0 and the SF-36 from weak to high [22,24,68]. These findings confirm that adults with a lower quality of life have a higher level of disability.

The total result and entire domains of the 36-item WHODAS 2.0 were statistically positively correlated with each domain of the HADS questionnaire; thus, a higher score on the 36-item WHODAS (higher disability) was associated with a higher score on the HADS questionnaire (anxiety and depression). These findings confirm that adults with higher anxiety and depression are characterized by a higher level of disability. We did not find any other studies in which the HADS scale was used for convergent validity for the 36-item WHODAS 2.0. However, studies using other scales assessing the occurrence of depression confirm the significant relationship between a higher level of depression and a higher level of disability. For instance, Rotarou et al. noticed a strong association of functional disability with increased depression in patients [69], whereas Sjonnesen indicated that the WHODAS 2.0 was sensitive in case of assessing the impact of depression [70].

The total result and all domains of the 36-item WHODAS 2.0 were significantly correlated with the ODI questionnaire. These findings confirm the hypothesis that adults with a higher physical disability measured by the ODI have a higher level of disability measured by the WHODAS 2.0. Saltychev et al. showed that the total scores of the WHODAS and the ODI were strongly correlated. Authors also suggested that the assessment of disability in the case of the population with LBP might be better estimated by the WHODAS 2.0 unlike the ODI [27]. According to Varjonen, both the WHODAS 2.0 and the ODI assessed the level of functioning of people experiencing LBP equivalently [28].

We confirmed that the 36-item WHODAS 2.0 had satisfactory validity for people with a different health status. In our study, the results of the WHODAS differed between people experiencing less and more pain. Similar situations were in all domains, except Do4 Getting along. These findings confirm our hypothesis that adults with higher levels of pain are characterized by a higher level of disability. With reference to the research performed by Baron et al., the patients with early arthritis were divided into two subgroups according to the results of the Center for Epidemiological Studies Depression Scale. These researchers noticed that the 36-item WHODAS 2.0 was able to distinguish patients with low and high depression symptoms [20]. Additionally, Garin et al. pointed out that as for most of the WHODAS 2.0 domains there were statistically significant, differences regarding groups with various clinical severity as for their medical condition and between professionally active and inactive due to poor health (*p* < 0.001) [30]. Serrano-Dueñas et al. revealed that in respect of the WHODAS 2.0, the getting along domain and the life activities domain were not significantly different in terms of staging according to the Hoehn and Yahrfor scale, whereas other domains and the total scale indicated differences [66].

The values of all WHODAS 2.0 domains changed between the first and third study. Results in all domains decreased significantly, i.e., the disability decreased. Almost all domains showed a moderate to large degree of responsiveness, respectively, for ES and SRM. The WHODAS 2.0 responsiveness scores for the total results were −1.35 (SRM) and −0.86 (ES) at 4 weeks after discharge. Meesters et al. found the WHODAS 2.0 responsiveness scores −0.35 (SRM), −0.34 (ES) at 6 weeks after discharge [71]. Moreover, in the research performed by Garin et al., analyzing the group of patients whose health condition had improved, they indicated small to moderate responsiveness coefficients (ES = 0.3–0.7), but higher than in the group of the SF-36 [30]. Additionally, Chwastiak et al. showed that the WHODAS 2.0 was well responsive (ES = 0.65) while assessing the treatment results [24].

Federici et al. emphasized that the 36-item WHODAS 2.0 is adequate for assessing disability and health status. Although it is an important issue for rehabilitation, MCID score for the WHODAS 2.0 should be established [19]. We found that the minimal clinically important difference in case of the total WHODAS 2.0 score in patients after rehabilitation for low back pain was 4.87. The largest MCID was found for Do2 Mobility (7.93 ± 0.70), and the smallest for Do1 Cognition (1.71 ± 0.347). The 36-item WHODAS can accurately capture changes in disability after rehabilitation in patients with LBP and thus can be used as a valid primary endpoint for clinical trials [72].

A weakness of our analysis is that our sample size prevented the use of a more robust test of the internal factor structure of the WHODAS 2.0 such as confirmatory factor analysis (CFA) [73]. The study’s strengths include the use of standardized methods for the assessment of psychometric properties. It is the first study in Poland and one of the few in the world to analyze the usefulness of the WHODAS 2.0 questionnaire in assessing the disability of patients with chronic back pain. The scientific foundation of this issue is particularly important in the context of the implementation of the ICF and the WHODAS 2.0 questionnaire in Poland for general use in rehabilitation departments and other physiotherapy units.

## 5. Conclusions

These findings show that the 36-item WHODAS is suitable for evaluating health status and disability, and that it is a reliable and valid tool for assessing patients with chronic LBP according to its psychometric properties. Because it can capture changes in disability after rehabilitation in patients with LBP, the 36-item WHODAS 2.0 can thus be used as a valid primary endpoint for clinical trials. Regarding that the 36-item WHODAS 2.0 is an easy-to-use, generic instrument, based on the principles of ICF, with high feasibility, it could be considered as a first-choice tool in the rehabilitation field that might implement the management of people with disability due to LBP.

## Figures and Tables

**Figure 1 ijerph-17-07284-f001:**
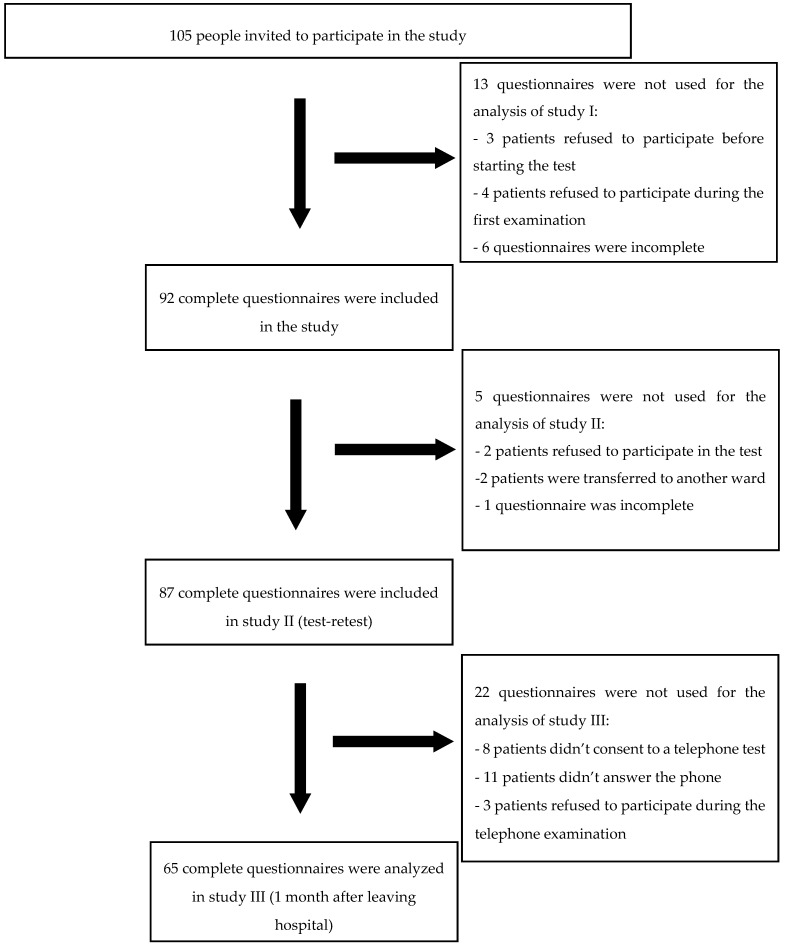
Flow diagram of the study.

**Table 1 ijerph-17-07284-t001:** General socio-demographic characteristics of the study population (n = 92).

Variables	Mean ± SD n (%)
1. Gender	
Female	57 (61.96)
Male	35 (38.04)
2. Age (years)	66.0 ± 11.6
3. Place of residence	
City	44 (47.82)
Countryside	48 (52.17)
4. Education	
Primary education	18 (19.57)
Vocational education	17 (18.48)
Secondary education	45 (48.91)
Higher education	12 (13.04)
5. Pain (VAS)	5.77 ± 1.31
6. 36-item WHODAS 2.0	41.53 ± 13.84
Do1 Cognition	15.98 ± 20.78
Do2 Mobility	65.08 ± 20.49
Do3 Self-care	34.13 ± 21.80
Do4 Getting along	23.37 ± 18.00
Do5 Life activities	60.43 ± 21.83
Do6 Participation	51.4 ± 18.17
7. SF-36	
Physical functioning	35.11 ± 20.91
Role physical	36.07 ± 19.87
Body pain	35.16 ± 17.06
General health	42.83 ± 14.92
Vitality	48.70 ± 15.37
Social functioning	57.34 ± 25.15
Role emotional	62.14 ± 28.26
Mental health	56.48 ± 13.89
PCS	37.14 ± 14.84
MCS	55.59 ± 14.55
8. ODI	29.57 ± 6.40
9. HADS	
Anxiety	7.67 ± 3.63
Depression	5.51 ± 3.14

**Table 2 ijerph-17-07284-t002:** Reliability of the 36-item WHODAS 2.0 for patients with LBP.

WHODAS 2.0	Floor Score	Ceiling Score	Cronbach’s α	ICC_2,1_ (95% CI)	SEM	MDC_95_
Do1 Cognition	35.87%	0.00%	0.896	0.899 (0.859–0.928)	6.48	17.96
Do2 Mobility	0.00%	7.61%	0.823	0.950 (0.93–0.965)	4.44	12.31
Do3 Self-care	8.70%	2.17%	0.815	0.805 (0.733–0.859)	10.06	27.88
Do4 Getting along	15.22%	0.00%	0.786	0.936 (0.910–0.955)	4.51	12.50
Do5 Life activities	0.00%	7.61%	0.904	0.759 (0.673–0.825)	11.01	30.52
Do6 Participation	0.00%	0.00%	0.830	0.897 (0.857–0.927)	5.61	15.55
Total score	0.00%	0.00%	0.921	0.928 (0.898–0.949)	3.77	10.45

**Table 3 ijerph-17-07284-t003:** Internal Structure of the 36-item WHODAS 2.0 for patients with LBP.

WHODAS 2.0 Domains	WHODAS 2.0 Total Score
	Pearson’s Correlation Coefficient
Do1 Cognition	r = 0.663, *p* < 0.001
Do2 Mobility	r = 0.762, *p* < 0.001
Do3 Self-care	r = 0.688, *p* < 0.001
Do4 Getting along	r = 0.482, *p* < 0.001
Do5 Life activities	r = 0.751, *p* < 0.001
Do6 Participation	r = 0.758, *p* < 0.001

**Table 4 ijerph-17-07284-t004:** Convergent validity of the 36-item WHODAS 2.0 for patients with LBP.

WHODAS 2.0		Do1 Cognition	Do2 Mobility	Do3 Self-Care	Do4 Getting Alone	Do5 Life Activities	Do6 Participation	Total Score
SF-36	Physical functioning	r = −0.388, *p* < 0.001	r = −0.784, *p* < 0.001	r = −0.502, *p* < 0.001	r = −0.312, *p* = 0.002	r = −0.647, *p* < 0.001	r = −0.666, *p* < 0.001	r = −0.810, *p* < 0.001
	Role physical	r = −0.125, *p* = 0.276	r = −0.554, *p* < 0.001	r = −0.362, *p* < 0.001	r = −0.133, *p* = 0.207	r = −0.540, *p* < 0.001	r = −0.551, *p* < 0.001	r = −0.539, *p* < 0.001
	Body pain	r = −0.466, *p* < 0.001	r = −0.784, *p* < 0.001	r = −0.525, *p* < 0.001	r = −0.389, *p* < 0.001	r = −0672, *p* < 0.001	r = −0.724, *p* < 0.001	r = −0.873, *p* < 0.001
	General health	r = −0.232, *p* = 0.026	r = −0.223, *p* = 0.033	r = −0.099, *p* = 0.350	r = −0.354, *p* < 0.001	r = −0.248, *p* = 0.017	r = −0.280, *p* = 0.007	r = −0.349, *p* = 0.001
	Vitality	r = −0.271, *p* = 0.009	r = −0.381, *p* < 0.001	r = −0.185, *p* = 0.078	r = −0.229, *p* = 0.028	r = −0.257, *p* = 0.014	r = −0.327, *p* = 0.001	r = −0.413, *p* < 0.001
	Social functioning	r = −0.260, *p* = 0.012	r = −0.477, *p* < 0.001	r = −0.075, *p* = 0.478	r = −0.010, *p* = 0.921	r = −0.325, *p* = 0.002	r = −0.579, *p* < 0.001	r = −0.476, *p* < 0.001
	Role emotional	r = −0.312, *p* = 0.003	r = −0.270, *p* = 0.009	r = −0.119, *p* = 0.257	r = −0.166, *p* = 0.113	r = −0.197, *p* = 0.059	r = −0.545, *p* < 0.001	r = −0.440, *p* < 0.001
	Mental health	r = −0.362, *p* < 0.001	r = −0.231, *p* = 0.027	r = −0.16, *p* = 0.126	r = −0.224, *p* = 0.032	r = −0.341, *p* = 0.001	r = −0.339, *p* = 0.001	r = −0.417, *p* < 0.001
	PCS	r = −0.395, *p* < 0.001	r = −0.781, *p* < 0.001	r = −0.495, *p* < 0.001	r = −0.361, *p* < 0.001	r = −0.681, *p* < 0.001	r = −0.705, *p* < 0.001	r = −0.834, *p* < 0.001
	MCS	r = −0.391, *p* < 0.001	r = −0.421, *p* < 0.001	r = −0.177, *p* = 0.091	r = −0.214, *p* = 0.041	r = −0.356, *p* < 0.001	r = −0.562, *p* < 0.001	r = −0.556, *p* < 0.001
HADS	Anxiety	r = 0.448, *p* < 0.001	r = 0.2, *p* = 0.046	r = 0.310, *p* = 0.003	r = 0.297, *p* = 0.004	r = 0.305, *p* = 0.003	r = 0.296, *p* = 0.004	r = 0.455, *p* < 0.001
	Depression	r = 0.294, *p* = 0.004	r = 0.358, *p* < 0.001	r = 0.319, *p* = 0.002	r = 0.360, *p* < 0.001	r = 0.398, *p* < 0.001	r = 0.342, *p* = 0.001	r = 0.489, *p* < 0.001
ODI		r = 0.585, *p* < 0.001	r = 0.667, *p* < 0.001	r = 0.526, *p* < 0.001	r = 0.345, *p* = 0.001	r = 0.645, *p* < 0.001	r = 0.716, *p* < 0.001	r = 0.867, *p* < 0.001

r from the Pearson’s correlation coefficient.

**Table 5 ijerph-17-07284-t005:** Known group validity of the 36-item WHODAS 2.0.

WHODAS 2.0		Pain (VAS Scale)		*p*-Value
		0 to 4	5 to 10	
Do1 Cognition	M ± SD	6.05 ± 9.51	18.56 ± 22.15	*p* = 0.018
	Median	0.00	10.00	
	Quartiles	0.00–10.00	0.00–30.00	
Do2 Mobility	M ± SD	44.74 ± 9.38	70.38 ± 19.26	*p* < 0.001
	Median	43.75	75.00	
	Quartiles	37.50–50.00	56.25–81.25	
Do3 Self-care	M ± SD	24.21 ± 16.10	36.71 ± 22.43	*p* = 0.004
	Median	20.00	30.00	
	Quartiles	20.00–30.00	20.00–50.00	
Do4 Getting along	M ± SD	18.42 ± 11.31	24.66 ± 19.22	*p* = 0.284
	Median	16.67	25.00	
	Quartiles	8.33–25.00	8.33–33.33	
Do5 Life activities	M ± SD	46.32 ± 10.65	64.11 ± 22.54	*p* = 0.001
	Median	40.00	60.00	
	Quartiles	40.00–50.00	50.00–90.00	
Do6 Participation	M ± SD	30.26 ± 9.81	56.91 ± 15.64	*p* < 0.001
	Median	29.17	56.25	
	Quartiles	25.00–37.50	45.83–66.67	
Total score	M ± SD	27.06 ± 6.85	45.29 ± 12.68	*p* < 0.001
	Median	25.00	43.48	
	Quartiles	23.91–28.26	35.87–52.17	

*p*-values from Student’s *t*-test.

**Table 6 ijerph-17-07284-t006:** Responsiveness of the 36-item WHODAS 2.0.

WHODAS 2.0	Change between 1^st^ and 3^rd^ Study			*p*-Value	SRM	ES	MCID	SE
	Mean	Median	SD					
Do1 Cognition	−3.69	0.00	8.49	*p* = 0.001	−0.43	−0.18	1.71	0.34
Do2 Mobility	−20.29	−18.75	18.75	*p* < 0.001	−1.08	−0.99	7.93	0.70
Do3 Self-care	−13.85	−10.00	14.76	*p* < 0.001	−0.94	−0.64	5.67	0.54
Do4 Getting along	−12.82	−8.33	15.46	*p* < 0.001	−0.83	−0.71	4.85	0.64
Do5 Life activities	−14.31	−10.00	17.76	*p* < 0.001	−0.81	−0.66	6.07	0.65
Do6 Participation	−11.15	−8.33	12.72	*p* < 0.001	−0.88	−0.61	4.62	0.47
Total score	−11.97	−11.96	8.86	*p* < 0.001	−1.35	−0.86	4.87	0.24

*p*-values from paired Student’s *t*-test.

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
