# Peer review of "Psychometric Properties of the Polish Version of the 36-Item WHODAS 2.0 in Patients with Low Back Pain"

_ijerph, 2020, doi:10.3390/ijerph17197284_

Round 1

Reviewer 1 Report

I think the authors have addressed the reviewers' comments satisfactorily.

Author Response

Dear Editor,

            I resubmit to you the second version of the manuscript entitled “Psychometric properties of the Polish version of the 36-item WHODAS 2.0 in patients with low back pain”. We are grateful to the editors and reviewers for your time and comments on our manuscript. Thank you for giving us the opportunity to revise again and resubmit this manuscript.

With kind regards,

Agnieszka Ćwirlej-Sozańska

Response to the review

Comments and Suggestions for Authors

I think the authors have addressed the reviewers' comments satisfactorily.

Thank you very much for Reviewer comment.

Reviewer 2 Report

NA. 

Author Response

Dear Editor,

            I resubmit to you the second version of the manuscript entitled “Psychometric properties of the Polish version of the 36-item WHODAS 2.0 in patients with low back pain”. We are grateful to the editors and reviewers for your time and comments on our manuscript.

With kind regards,

Agnieszka Ćwirlej-Sozańska

Response to the review

Comments and Suggestions for Authors

NA

Thank you for accepting the changes we have made to the manuscript.

Reviewer 3 Report

Review 1
Comments and Suggestions for Authors
This is a clearly written paper offering sound advice for those wishing to assess level of reported disability among patients with chronic lower back pain.
However, I wonder why the authors do not utilize a confirmatory factor analysis (CFA) to analyze the internal structure of the WHODAS 2.0 to test whether scale items in fact align with intended constructs among this sample?
Answer:
We do not utilize a confirmatory factor analysis (CFA) because of a statistical fact, that a minimum population of 500 is needed for CFA [1].
1. Booth T, Hughes DJ. The Wiley Handbook of Psychometric Testing: A Multidisciplinary Reference on
survey scale, and test development. Volume 1. Edited by Paul Irwing, Hoboken NJ USA, Chichester West
Sussex UK, Wiley Blackwell 2018

suggestion: It would be most helpful to cite (page numbers) where in the text the minimum size of 500 required for CFA is described in the book you included. There is a hearty discussion on sample size with regard to CFA, but a conclusive minimum number is not clearly agreed upon. I’ve most often seen recommendations for 5 or 10 (Nunnally & Bernstein, 1967; Wang & Wang, 2012) cases per indicator variable or parameter (Bentler & Chou, 1987), but advances with Monte Carlo simulations seem to be leading to the constant questioning and revising of recommendations.

This study performed all appropriate clinimetric analysis it could from the data available. However, the sample used in this study provided input to another, multi-site, pooled data study involving patients with various musculoskeletal problems that performs CFA.
This Polish study providing an new tool that can be used and considered by clinicians and researchers then further analysed for the finer and more complicated analysis within the total spectrum of required realms of the full standard and clinimetric requirements – both the psychometric and the practical.
We would like to mention, that the NDI and ODI are the most commonly used neck and low back PROs and were developed and first published around 40 years ago – however neither underwent CFA analysis respectively till 2014-16 and 2017, both through pooled samples of international data in which some of this article’s authors were involved (During that 40 years each tool was translated into more than 15-20 languages yet not one of these published cultural and translation studies performed CFA). Instead of CFA, we performed the internal structure study in this group, which, to some extent, replaces the CFA analysis.

suggestion: Thank you for sharing. The future CFA study appears to have great potential to contribute to research and practice. It is interesting to learn more of the history of similar tools. While I do not agree that examining correlations between scale items and subconstructs or overall scores is comparable to a CFA, I am satisfied with the justification you provide with a slight revision to the statement you added on page 13, line 449. Please consider editing to read:

A weakness of our analysis is that our sample size prevented the use of a more robust test of the internal factor structure of the WHODAS2.0 such as confirmatory factor analysis (CFA).

The manuscript has been proofread by a native English speaker.

Author Response

Dear Editor,

            I resubmit to you the second version of the manuscript entitled “Psychometric properties of the Polish version of the 36-item WHODAS 2.0 in patients with low back pain”. We are grateful to the editors and reviewers for your time and comments on our manuscript. Thank you for giving us the opportunity to revise again and resubmit this manuscript.

            According to the comments raised by the reviewers, we modified our manuscript.

            We are now resubmitting the revised manuscript and also our response to the comments. All changes are highlighted as red text in the manuscript.

Many thanks for the chance of improving.

            We hope you will be pleased with the changes, and support the publication of our revised manuscript.

With kind regards,

Agnieszka Ćwirlej-Sozańska

Response to the review

Comments and Suggestions for Authors

This is a clearly written paper offering sound advice for those wishing to assess level of reported disability among patients with chronic lower back pain.

However, I wonder why the authors do not utilize a confirmatory factor analysis (CFA) to analyze the internal structure of the WHODAS 2.0 to test whether scale items in fact align with intended constructs among this sample?

Answer:

We do not utilize a confirmatory factor analysis (CFA) because of a statistical fact, that a minimum population of 500 is needed for CFA [1].

  1. Booth T, Hughes DJ. The Wiley Handbook of Psychometric Testing: A Multidisciplinary Reference on survey scale, and test development. Volume 1. Edited by Paul Irwing, Hoboken NJ USA, Chichester West Sussex UK, Wiley Blackwell 2018 suggestion:

It would be most helpful to cite (page numbers) where in the text the minimum size of 500 required for CFA is described in the book you included. There is a hearty discussion on sample size with regard to CFA, but a conclusive minimum number is not clearly agreed upon. I’ve most often seen recommendations for 5 or 10 (Nunnally & Bernstein, 1967; Wang & Wang, 2012) cases per indicator variable or parameter (Bentler & Chou, 1987), but advances with Monte Carlo simulations seem to be leading to the constant questioning and revising of recommendations.

This study performed all appropriate clinimetric analysis it could from the data available. However, the sample used in this study provided input to another, multi-site, pooled data study involving patients with various musculoskeletal problems that performs CFA.

This Polish study providing an new tool that can be used and considered by clinicians and researchers then further analysed for the finer and more complicated analysis within the total spectrum of required realms of the full standard and clinimetric requirements – both the psychometric and the practical.

We would like to mention, that the NDI and ODI are the most commonly used neck and low back PROs and were developed and first published around 40 years ago – however neither underwent CFA analysis respectively till 2014-16 and 2017, both through pooled samples of international data in which some of this article’s authors were involved (During that 40 years each tool was translated into more than 15-20 languages yet not one of these published cultural and translation studies performed CFA). Instead of CFA, we performed the internal structure study in this group, which, to some extent, replaces the CFA analysis.

suggestion: Thank you for sharing. The future CFA study appears to have great potential to contribute to research and practice. It is interesting to learn more of the history of similar tools. While I do not agree that examining correlations between scale items and subconstructs or overall scores is comparable to a CFA, I am satisfied with the justification you provide with a slight revision to the statement you added on page 13, line 449. Please consider editing to read:

A weakness of our analysis is that our sample size prevented the use of a more robust test of the internal factor structure of the wHODAS2.0 such as confirmatory factor analysis (CFA).

The manuscript has been proofread by a native English speaker.

Response:

Thank you very much for Reviewer comment. It is very valuable to us. We are aware of the discussion on the size of the group needed to perform a reliable CFA analysis. We also know that the proposed study of the internal structure is only a substitute. According to the information provided, we plan to perform CFA for a larger group of patients with various musculoskeletal problems. To the limitations in the manuscript, we added the sentence proposed by the Reviewer and a reference with the relevant pages. Thanks again for your kindness and support.

This manuscript is a resubmission of an earlier submission. The following is a list of the peer review reports and author responses from that submission.

Round 1

Reviewer 1 Report

This is a clearly written paper offering sound advice for those wishing to assess level of reported disability among patients with chronic lower back pain. 

However, I wonder why the authors do not utilize a confirmatory factor analysis (CFA) to analyze the internal structure of the WHODAS 2.0 to test whether scale items in fact align with intended constructs among this sample?

Reviewer 2 Report

Introduction  Page 2, line 56: Change “experience” to “experienced”  Page 2, line 64: Please clarify/rework the sentence, “However, data regarding the validity of the 36-item WHODAS 2.0 in 64 chronic low back pain in Poland.”

Materials and Methods  Page 2, line 81: Change “study 1” to “study I” and change “months” to “month”  Page 2, line 82: Change “87 – from” to “87 from study II” Page 2, line 83: Change “65 – third” to “65 from study III”  Figure 1:  Change the word “refusal” to “refused”.  Change, “87 complete questionnaires were included in the study II (test-retest)” to “87 complete questionnaires were included in study II (test-retest)”.  The sentence, “3 patients refused to participate during…” is not complete – please clarify. Change “65 complete questionnaires were analyzed in the study III (1 month after leaving the hospital)” to “65 complete questionnaires were analyzed in study III (1 month after leaving the hospital)”.  Post-hoc power analyses provide little useful information. I therefore recommend Section 2.2 Sample Size and Table 1 be left out of the paper.

Measurements.  Page 4, line 132: Rewrite to read as, “ The domains included: Domain 1 (Do1) Understanding…” Page 5, line 183: Because this is a descriptive analysis of the psychometric properties of the WHODAS 2.0, statistical significance is not needed and the sentence, “The statistical significance was determined on the level of 183 p < 0.05” can be deleted.

Reliability.  Page 5, line 188: Change to “… adequate internal consistency reliability…” 

Test-retest reliability:  Replace “tool” with the name of the scale. Reliability is consistency and so the following sentence is not clear: “During this period there should be significant changes in the phenomenon under study.” If the scale is reliable, one would expect little change, particularly over just a 2-day interval.  Clarify the 2-day interval with respect to the studies - the 2-day interval was the interval between study I and study II - correct? Please indicate which ICC model was used for the analysis. Please briefly define relative and absolute reliability.  

Internal structure: Remove the clause, “…and the results were supplemented with the results of the significance of the correlation coefficient test.”

Validity.  Convergent validity: Please remove the word “hypothesized” and change the sentence to read something like, “Adults with a lower quality of life and lower mood as measured by the _________ should have a higher level of disability as assessed with the ___________ [44]. Since this is not a hypothesis-driven study, but a descriptive analysis of the psychometric properties of a scale, hypotheses per se are not necessary.

Known-group validity: Line 211: delete the clause, “according to the hypothesis”. Line 227: Change to, “Adults with higher levels of pain should be characterized by a higher level of disability.”

Responsiveness: Please define responsiveness. Is responsiveness part of validity or is it a separate measure?  If responsiveness is a separate concept, it should have a different heading not located under validity. Three different statistics are described to assess responsiveness. This is OK, but the definitions of these statistics are not clear. For example, “ES is defined as a score change in the 36-item WHODAS 2.0 (between 1 and 3 test) divided by baseline SD” is not clear because “score change” is ambiguous – is score change a change in score for a individual subject or the average for the group? Please also add that a paired-samples t-test was used to examine the mean change between test 1 and test 3.  Overall, responsiveness  is not clear and must be reworked.

Results. Socio-demographic characteristics:  Table 1: I might change Mean ± SD / n (%) to Mean ± SD or n (%) so the / does not get confused with a mathematical symbol.

Reliability.  Internal consistency Line 257: Replace “tool” with the name of the instrument.

Test-retest reliability and measurement error: Line 263: Replace “over time” with “over an approximate 2-day period”. Line 268: Please interpret the meaning of the MDC of 10.45.  The sentence could read as, “ The Minimal Detectable Change was the best (the smallest) for the total result (MDC95= 10.45) indicating that...”  Line 270: What is “L-S” problem?  I believe this represents Lumbar–Sciatica. Please clarify for readers. I do not see that a test-retest was conducted for the longer interval (between study II and III) - please explain why.

Internal structure: Table 3: Delete the asterisks (*) in the table and *Statistically significant (p < 0.05) below the table.

Floor and ceiling effects: Rephase to read as, “However, over 15% of respondents reported the lowest possible score for WHODAS 2.0 Do1 Cognition and Do4 Getting along, indicating possible floor effects for these two domains. (Table 2).

Validity. Convergent validity: Line 293-294: Deleted the first sentence “All correlation coefficients were statistically significant at the level of p<0.05” and reword the second sentence to read as, “These findings confirm that adults with a higher anxiety and depression are characterized by a higher level of disability (Table 4).” Lines 296-298:  Rephase to read as, “The total result of the 36-item WHODAS 2.0 and all domains were correlated with the ODI questionnaire. These findings confirming that adults with a higher physical disability measured by the ODI have a higher level of disability by the 36-item WHODAS 2.0 (Table 4).” Table 4: Delete the asterisks (*) in the table, delete *Statistically significant (p < 0.05) below the table, and change “r – Pearson’s correlation coefficient” to “r is Pearson’s correlation coefficient.”

Known Group Validity: Rephrase to read as, “With the possible exception of Do4 Getting along, we found probable differences between the selected subgroups and the total score of the 36-item WHODAS 2.0. These findings indicate that adults with higher levels of pain are likely characterized by a higher level of disability (Table 5).” Table 5: In the column headings, change “points from 0 to 4” to “0 to 4” and “points from 5 to 10” to “5 to 10”. Delete the asterisks (*) in the table and change “*p - Student's t-test below the table to read as, “p-values from Student’s t-test”. Change “Mean” to “M”. Reposition the p-values in the column so they line up with the M ± SD instead of the Median.

Responsiveness: Rephrase to read as, “The statistical evidence indicates that all WHODAS 2.0 domains decreased between the first and third study (i.e., from test 1 to test 3). Nearly all domains showed a moderate to large degree of responsiveness, respectively, as signified by ES and SRM values. The largest MCID was found in the case of Do2 Mobility (7.93 ± 0.70), and the smallest in the case of Do1 Cognition (1.71 ± 0.347) (Table 6).”  Table 6: In the column heading revise “Change test 3 vs test 1” to read as “Change: test 1 minus test 3”. Delete the asterisks (*) in the table. Delete *Statistically significant (p < 0.05) below the table and change p – paired Student's t-test below the table to read as, p-values from paired Student's t-test.

Discussion:  Page 12, line 386: Delete the word “significantly”.  Line 393: Delete “the hypothesis”. Line 398: delete “the hypothesis”. Line 401: Delete “significant”. Line 406: Delete “significantly”. Line 407: Delete “the hypothesis”.  Line 414: Delete “significantly”. Delete all subsequent occurrences of these phrases. Page 14, Line 445: Change “demonstrated” to “show”.  General comment. The Discussion should contain most of the following elements:  Restate the purpose and summarize the results; restate each aim and discuss how the data fulfilled that aim and if the results agree or do not agree with previously cited literature in the Introduction; discuss theoretical and practical implications of the results (if possible); identify study strengths and limitations; provide recommendations for future research; summarize and state conclusions.  Please check for these items in the Discussion.  From my reading of the Discussion, all items appear to be present except for a summary of the limitations and strengths of the study.

Reviewer 3 Report

Overall speaking, the paper is quite interesting. The study described is novel in the Polish context but the issue has been extensively discussed elsewhere.  Therefore, I would suggest the authors to conduct a more comprehensive literature review on similar previous studies.  It would be meaningful if these previous findings can be compared with the findings of the current study.

However, I have to say that the paper must be rejected due to serious problem of plagiarism.  As shown in the attached iThenticate report, the similarity index is 33%. It is unacceptably high, particularly for a quantitative research paper.

Apart from these comments, the authors should address or note the following issues:

1) The use of language should be improved. There are quite many typos and grammatical errors (though minor). For example, a fullstop is missed in the abstract (line 28).

2) The structure of the paper is awful. In particular, in the introduction section, one-sentence paragraph exists.

3) Moreover, the shortforms should be explained where they are first used.  ICC is one of the examples.  It is used in page 3 but its explanation appears not until page 5.

4) The sampling method is not well explained. In line 76-77, the authors described the qualifying criteria only. Does it mean that all patients qualified were invited to complete the survey?

5) What scales were used in different stages of the study?
